# Asset Management Method of Industrial IoT Systems for Cyber-Security Countermeasures

**Noritaka Matsumoto** [1,*]**, Junya Fujita** [1]**, Hiromichi Endoh** [1]**, Tsutomu Yamada** [2]**, Kenji Sawada** [3] **and Osamu Kaneko** [3]

1 Hitachi, Ltd. Research & Development Group, 7-1-1 Omika-cho, Hitachi, Ibaraki 319-1292, Japan; junya.fujita.so@hitachi.com (J.F.); hiromichi.endo.be@hitachi.com (H.E.)

2 Hitachi, Ltd. Service & Platform Business Unit, 5-2-1 Omika-cho, Hitachi, Ibaraki 312-1293, Japan; tsutomu.yamada.bs@hitachi.com

3 Department of Mechanical and Intelligent Systems Engineering, The University of Electro-Communications, 1-5-1 Chofugaoka, Chofu, Tokyo 102-0076, Japan; knj.sawada@uec.ac.jp (K.S.); o.kaneko@uec.ac.jp (O.K.)

\* Correspondence: noritaka.matsumoto.ak@hitachi.com

**Abstract:** Cyber-security countermeasures are important for IIoT (industrial Internet of things) systems in which IT (information technology) and OT (operational technology) are integrated. The appropriate asset management is the key to creating strong security systems to protect from various cyber threats. However, the timely and coherent asset management methods used for conventional IT systems are difficult to be implemented for IIoT systems. This is because these systems are composed of various network protocols, various devices, and open technologies. Besides, it is necessary to guarantee reliable and real-time control and save CPU and memory usage for legacy OT devices. In this study, therefore, (1) we model various asset configurations for IIoT systems and design a data structure based on SCAP (Security Content Automation Protocol). (2) We design the functions to automatically acquire the detailed information from edge devices by "asset configuration management agent", which ensures a low processing load. (3) We implement the proposed asset management system to real edge devices and evaluate the functions. Our contribution is to automate the asset management method that is valid for the cyber security countermeasures in the IIoT systems.

**Keywords:** control system security; IIoT; asset management; SCAP; SysML

## 1. Introduction

The term IIoT (industrial IoT) refers to industrial systems with advanced IoT technology. With these systems, the number of devices connected to networks is increasing, and steps are also being taken to integrate different types of systems and support open/multivendor architectures. IIoT technology uses a diverse variety of wired and wireless connection methods, which makes the network configuration more complex and makes it difficult to manage devices manually. As of 2020, the Ministry of Internal Affairs and Communications estimates that the number of IoT devices for industrial applications, such as automation and energy, will increase at an annual rate of approximately 20%, reaching about 9.27 billion units worldwide by 2022 [1].

The increase in IoT devices will further accelerate the incidence of cybersecurity issues in social infrastructure systems. In fact, there have been cases where vulnerabilities in numerous IoT devices have been exploited by malware targeting equipment, such as security cameras and home routers [2]. The implementation of security countermeasures of this sort in infrastructure systems is a problem due to the increased expenditure of time and money on introducing countermeasures to all IoT equipment in the system. To respond promptly, it is important to have an accurate understanding of the IoT devices from which a system is configured, conduct a system risk analysis, and manage the risk by introducing countermeasure technologies starting with the IoT devices that are most at risk [3].

In this paper, we propose a method for streamlining the configuration and management of equipment (assets) in order to contribute to the automation of the security measures in IIoT systems, which have issues peculiar to control systems. As described in Section 3, these control system issues include the necessity of supporting a wide variety of devices and network configurations while satisfying the following requirements:

(a)　Real-time performance

In general, there must be no disruption of the real-time communication and control processes with a 10–100 ms cycle duration. This makes it difficult to install security countermeasures, such as the malware detection software used in IT systems.

(b)　Frugal resource usage

Systems must have resource-saving designs that are compatible with legacy equipment. There is still a lot of legacy equipment in control systems that use devices with CPU operating frequencies on the order of MHz and memory capacities on the order of kilobytes to megabytes, which makes it difficult to introduce software that presupposes the availability of abundant computing resources (CPUs running at GHz frequencies with gigabytes of memory).

(c)　High reliability and high availability

The high reliability and high availability of the equipment in the field must be guaranteed. The control systems must operate continuously for 24 h a day, 365 days a year, with a service life of at least 5–10 years. It is, therefore, not acceptable to carry out security updates that require a system to be stopped or restarted, as is the case with IT systems. It is also unacceptable for security updates to cause system malfunctions [4,5].

To solve these problems, we propose an asset configuration management method that can support the configuration of the systems in industrial fields where various networks and devices are used. Specifically, this method involves modeling the IIoT system and then specifying the data structures suitable for managing its asset configuration. We also propose a method for collecting and managing the asset information in a system that is useful for security measures [6] while satisfying the above requirements that are specific to control systems (i.e., (a) real-time performance, (b) frugal resource usage, and (c) high reliability and high availability). We have designed and prototyped an agent for this purpose (called an asset configuration management agent), and, in evaluations using actual machines, we have shown that it can improve the efficiency of asset configuration management in IIoT systems.

This paper is organized as follows: Section 2 discusses related research about the cyber security [7–10], automation [11–13], and asset management [14–17] for IIoT and OT systems. Section 3 proposes the modeling method and data structure for IIoT systems, and Section 4 designs the asset configuration management method. Section 5 evaluates the prototyping of the asset configuration management agent, and Section 6 describes conclusions and future works. Our contributions are as follows. (1) We have modeled the configuration of various control systems and designed the structure of asset configuration management data that should be collected for security countermeasures. (2) We have proposed a method to automate the asset configuration management by using an agent for IIoT systems. (3) We have developed and evaluated the prototype of an asset configuration management agent that can execute its functional processing without disturbing real-time control.

## 2. Related Research

We investigate the existing studies related to cyber security issues in IIoT and OT systems. The studies are categorized by advanced security taxonomy for machine-to-machine communications [7], cyber threats and standards landscape for IIoT [8], trust management, privacy, authentication, threats and access control [9], and secure location-based authentication scheme in fog computing environments [10]. In addition, we refer to studies related to automation in IIoT and OT systems below. There are surveys about

process automation in an IoT–fog–cloud ecosystem [11], a framework of automation on context-aware IoT systems [12], and a real-time plant environment simulator [13].

Studies related to the asset configuration management method proposed in this paper are summarized below. In Japan, an independent administrative agency called the Information-technology Promotion Agency (IPA) has published a set of asset management guidelines for control systems [14] for the purpose of dealing with cybersecurity risks in control systems. This document highlights the following five issues that concern asset management in control systems: (a) asset management is not performed from a security perspective. (b) It requires a lot of time and effort to implement. (c) People do not know what standard to follow with regard to the extent of asset management. (d) There are concerns that asset management measures are insufficiently comprehensive. (e) Since asset management is performed on an ad hoc basis, the operational rules that people should follow are either impossible to find or non-existent. Furthermore, we organize the asset information that should be collected in control systems with respect to the relationship between asset information and threats. We also verify the methods for collecting asset information (including the use of management systems) and the tools for the automation of this process.

Other studies on asset management include the proposal of an asset container method using the CWSS (Common Weakness Scoring System) and CVSS (Common Vulnerability Scoring System) [15], a study relating to a control system asset management tool that takes security into consideration [16], and an interface for the standardized management of control system assets (AAS: Asset Administration Shell) [17].

The asset configuration management method proposed in this paper differs from these related studies in that it supports automation in the diverse system configurations exhibited by control systems and satisfies the specific requirements of control systems ((a) real-time performance, (b) frugal resource usage, (c) high reliability and high availability) while collecting information that is essential from the viewpoint of security measures.

## 3. Model and Data Structure of IIoT Systems

In this section, we propose the modeling method and data structure for IIoT systems. We use SysML to model the configuration of a general control system and the functions of an asset configuration management agent, and we apply CPE to define the asset configuration management data structures to be collected for security measures, and to facilitate linkage with IT systems.

### 3.1. Modeling of IIoT Systems

Since IIoT systems consist of a wide variety of devices and networks, it is important to accurately grasp and manage their asset structure. To prepare for understanding the asset structure of IIoT systems, we decided to analyze the existing control systems and model their assets.

To model these asset configurations, we are investigating the configuration of the ordinary systems used for thermal power generation, water treatment plants, chemical plants, manufacturing plants, building control systems, and power transmission/distribution systems, and we are analyzing what types of devices they use, in what numbers, how these devices are interconnected, and so on. We classified the asset configuration into hardware, software, and network assets, and we are studying the equipment composition of each asset class. During our survey, we noticed that, although there are differences in the detailed configuration of all the control systems, their basic configuration follows the reference model defined in ANSI/ISA95 [18] (Figure 1).

The international standards regarding industrial control systems, such as ISA99 and IEC62443, also refer to the hierarchical structure according to the ISA95 model [19,20]. In considering the security of the IIoT system, it is important to consider the system configuration that combined the IT and OT targeted by ISA99 and IEC62443 [21]. In this paper, we will focus on the OT side, where there are still many security challenges left. Therefore, we

are particularly concerned with the hardware, software, and networks that constitute Level 2 (supervisory control systems), Level 1 (basic control devices), and Level 0 (field devices). The principal asset components are hardware assets, such as controllers, servers, and field devices. There are also software assets, such as operating systems and applications, installed on these hardware assets, and network assets that interconnect the hardware assets. Based on the results of the abovementioned analysis of asset configurations in diverse systems, we formulate a method for modeling the asset configurations of IIoT systems in a common modeling language called Systems Modeling Language (SysML) [22], whose specifications were developed by the OMG (Object Management Group). SysML can be used for the overall system design of hardware and software (specification, requirements analysis, design, verification, etc.). Using a SysML block definition diagram, we create a reference architecture (Figure 2) that is both versatile and extensible and is capable of representing the asset structure of an IIoT system. Based on this reference architecture, system modeling and asset configuration management are performed according to the asset configuration of the actual IIoT system.

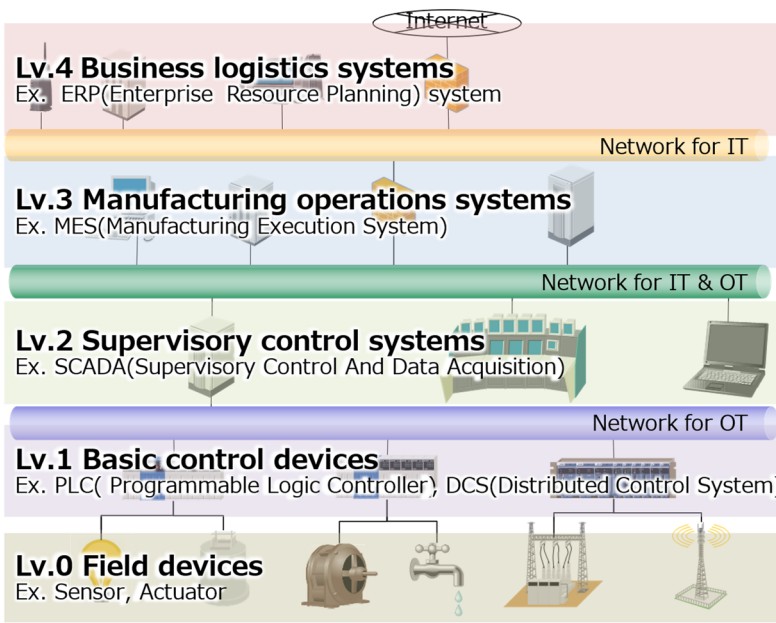

**Figure 1.** ANSI/ISA95 reference model used for ISA99 and IEC62443.

The procedure for creating a reference architecture is as follows. First, the assets of the IIoT system are described as SysML blocks and interfaces. Specifically, the hardware assets (e.g., PCs and controllers) and software assets (e.g., operating systems and applications installed on them) are described as blocks, and the networks that connect them together are described as interfaces. In addition, the blocks and interfaces are connected to each other by the three notation methods of association, generalization (inheritance), and aggregation prescribed by the SysML specification. Association indicates that blocks and interfaces are related (connected) to each other. Generalization indicates that one block's functionality is inherited by another block. Aggregation indicates that one block is a component of a set that includes one or more other blocks.

Figure 2 shows that the client PC block is connected to the controller block via the industrial network interface. In addition, the PLC block inherits the functions of the controller block. This means that it is a specific type of controller, for example, a PLC (programmable logic controller) or DCS (distributed control system). The figure also shows that the operating system (OS) block is a component (aggregate) element of the PLC block.

Versatile modeling can be performed by referring to IEC61784 [23] and IEC61158 [24], which have been standardized for industrial networks. For example, IEC61784 specifies network types, such as Fieldbus profiles (Part 1), Real-time Ethernet (Part 2), and functional

safety fieldbuses (Part 3). Furthermore, IEC61784 Part 1 defines multiple network protocols in the form of CPFs (Communication Profile Families), such as CPF1 (FOUNDATION Fieldbus). Therefore, if the interfaces are modeled according to IEC61784, which is inherited from industrial networks, IEC61784-Part 1, which inherits from IEC61784, and CPF1 (FOUNDATION Fieldbus), which inherits from IEC61784-Part 1, then it will be highly versatile and extendable.

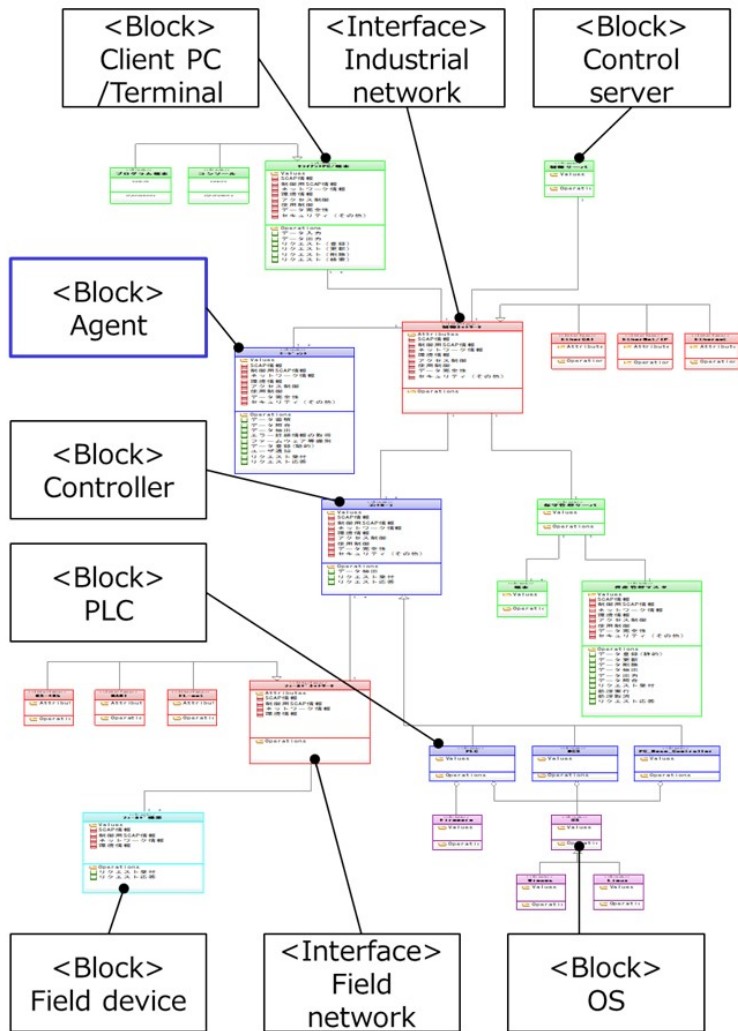

**Figure 2.** Reference architecture.

An asset configuration management agent (described later) is also added to this reference architecture as an agent block, whose role is to collect asset configuration management data in the system. In Figure 2, the asset configuration management agent is shown as an external function that operates independently of the controller, but it could also be a constituent software element of the controller.

### 3.2. Specification of Asset Configuration Management Data Structure

In the IIoT system modeled in the previous section, we specified a suitable data structure whereby the asset configuration management agent can efficiently collect and manage information that is effective for security measures. For this purpose, we classified the devices into ordinary IoT equipment and so-called operational technology (OT) devices, and we organized the information that can be acquired from each device class. We then studied data structures that can be used to identify security vulnerabilities based on the information acquired from each device. The data structure we specify is shown in Table 1.

**Table 1.** Data structure for asset configuration management.

| Classification of IIoT Device | Information for Device Management | | | | | | | | | | | | | | | | | | | | | | | |
|---|---|---|---|---|---|---|---|---|---|---|---|---|---|---|---|---|---|---|---|---|---|---|---|---|
| | Information Based on SCAP CPE | | | | | | | | | | | | | | | Extended Information | | | | | | | | |
| | SCAP Information | | | | | | | | | | SCAP Info. (Extended for IIoT) | | | | | Device Information | | Network Information | | | | | | |
| | Part | Vendor | Product | Version | Update | Edition | Software Edition | Target Software | Target Hardware | Language | Device Class1 | Device Class2 | Device Class3 | Target Network | ID/Product Number | Location | Extend Information | Time Stamp | Network Type | Physical Address | Network Address | Neighbor Device 1 | ... | Neighbor Device N |
| Client | ○ | ○ | ○ | ○ | ○ | - | ○ | - | - | ○ | ○ | ○ | △ | ○ | ○ | ○ | - | ○ | △ | ○ | ○ | - | - | - |
| Mobile | ○ | ○ | ○ | ○ | ○ | - | ○ | - | - | ○ | ○ | △ | △ | ○ | ○ | ○ | - | ○ | △ | ○ | ○ | - | - | - |
| Server | ○ | ○ | ○ | ○ | ○ | - | ○ | - | - | ○ | ○ | △ | △ | ○ | ○ | ○ | - | ○ | △ | ○ | ○ | - | - | - |
| Network | ○ | ○ | ○ | ○ | ○ | - | ○ | - | - | - | ○ | ○ | △ | ○ | ○ | ○ | - | ○ | △ | ○ | ○ | △ | △ | △ |
| IoT device | ○ | ○ | ○ | ○ | ○ | - | ○ | △ | △ | - | ○ | ○ | △ | ○ | ○ | ○ | - | ○ | △ | ○ | ○ | △ | △ | △ |
| Controller | ○ | ○ | ○ | ○ | ○ | - | - | ○ | ○ | ○ | ○ | △ | △ | ○ | ○ | ○ | - | ○ | △ | ○ | ○ | △ | △ | △ |
| HMI | ○ | ○ | ○ | ○ | ○ | - | - | - | △ | - | ○ | △ | △ | ○ | ○ | △ | - | △ | △ | △ | △ | △ | △ | △ |
| Sensor | ○ | ○ | ○ | ○ | - | - | - | - | △ | - | ○ | △ | △ | ○ | ○ | △ | - | - | △ | △ | △ | △ | △ | △ |
| Switch | ○ | ○ | ○ | ○ | - | - | - | - | △ | - | ○ | △ | △ | ○ | ○ | △ | - | - | △ | △ | △ | △ | △ | △ |
| Actuator | ○ | ○ | ○ | ○ | ○ | - | - | - | △ | - | ○ | △ | △ | ○ | ○ | △ | - | △ | △ | △ | △ | △ | △ | △ |
| Power | ○ | ○ | ○ | ○ | - | - | - | - | △ | - | ○ | △ | △ | ○ | ○ | △ | - | - | △ | △ | △ | △ | △ | △ |
| I/O device | ○ | ○ | ○ | ○ | ○ | - | - | - | △ | - | ○ | △ | △ | ○ | ○ | △ | - | - | △ | △ | △ | △ | △ | △ |
| Analyzer | ○ | ○ | ○ | ○ | ○ | - | - | - | △ | - | ○ | △ | △ | ○ | ○ | △ | - | △ | △ | △ | △ | △ | △ | △ |
| Recorder | ○ | ○ | ○ | ○ | ○ | - | - | - | △ | - | ○ | △ | △ | ○ | ○ | △ | - | - | △ | △ | △ | △ | △ | △ |

The rows in this data structure correspond to different classifications of IIoT devices (e.g., controllers, network devices, IoT devices, actuators, sensors, I/O devices, and switches), and the columns correspond to different types of information to be obtained for device management. The ability to automatically acquire each type of information from a device is represented by one of the following three symbols: ○: possible, △: possible under specific conditions, and –: difficult/unknown.

We specified a data format for describing specific asset information in this data structure. To create a highly versatile data format, we designed a format that is compatible with CPE (Common Platform Enumeration) [25] in the specification of SCAP (Security Content Automation Protocol) [26], which is a standard used in the management of IT system assets. In CPE, the device asset information is described in XML format using the Applicability Language specification. The asset information is also managed as a dictionary database according to the Dictionary specification. The methods used to link a name to each asset are specified by the Naming specification and the Name matching specification. For details of how SCAP/CPE for IT systems is applied to control systems, please refer to our previous work [27]. The use of CPE for IIoT asset configuration management makes it easy establish links with the asset configuration management of existing IT systems.

With information in CPE format as basic information, we specify extended information, including network information and device information, such as the ID/manufacture numbers for solid identification of assets, installation location, and time stamp of information acquisition. For the network information, we considered that it is important to acquire information such as the corresponding network type, physical address, logical address information, and information on adjacent connected devices.

Figure 3 shows an example where a specific IIoT system configuration is described in an XML data format compatible with the CPE specified in this paper. The system consists of a platform and individual assets. A platform is a unit of asset information that comprises a combination of individual assets. For example, a controller platform might consist of a CPU module, various network modules, and an I/O module. Each module consists of a platform with individual assets, such as a CPU and an operating system. According to this configuration information, we generate a list of each individual asset and a list of the platforms, which are used in asset configuration management for security measures.

As described above, to manage the diverse variety of asset information used in IIoT systems, this paper specifies a data structure that can be managed in a format compatible with the CPE specification. As a result, CPE databases (dictionary information) that have been accumulated in the IT systems can also be used in the IoT and OT systems. Conversely, the CPE information specific to the IoT and OT systems can also be reflected and stored in the IT systems.

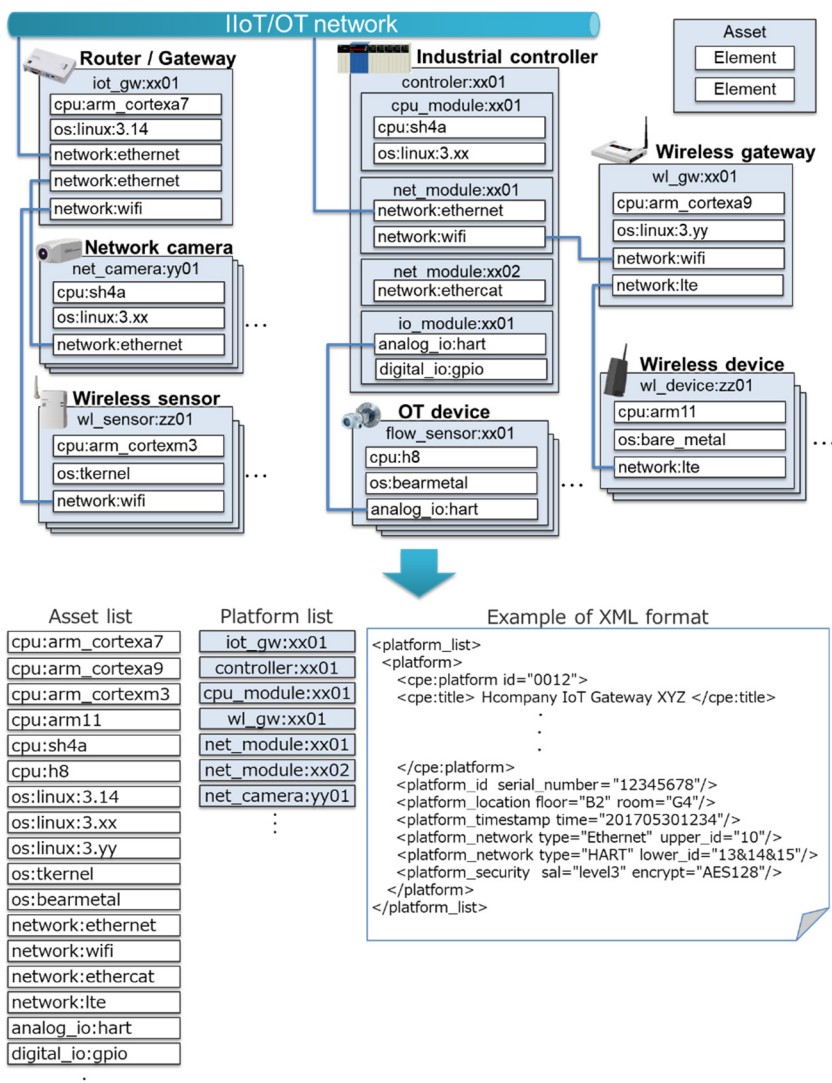

**Figure 3.** Example of IIoT system structure.

### 4. Asset Configuration Management Method for IIoT Systems

This section discusses a method for automating the asset configuration management by the asset configuration management agent to collect essential information for security countermeasures. As concrete examples of how asset configuration management can be achieved without interfering with real-time control, we design functions to acquire version information from installed software and check system passwords. Our targets are set so that the agent functions will cause the CPU load factor and memory occupancy to increase by no more than 1%.

#### 4.1. Selection of Asset Configuration Management Information to Be Collected

Considering the control system-specific requirements described in Section 1 ((a) real-time performance, (b) frugal resource usage, and (c) high reliability and high availability), we select the information that should be prioritized for collection by the asset configuration management agent in the IIoT system. To implement effective security measures, it is important to collect the following information on IIoT devices:

(d) Individual identification information: manufacturer's information, serial number, type of device, etc.

(e) Network connection information: type of network protocol used, communication port, address setting, etc.

(f)  Software configuration information: version information of installed software, patched status, etc.

(g)  Information on security settings: execution and access privileges of software installed on the device, system password information, etc.

IIoT devices carry various types of information of this sort, and the information that can be acquired differs depending on the type of operating system installed on the device. In this paper, we considered the following two types of information acquired by the asset configuration management agent to be of the highest priority.

(1)  Software version information

In principle, the application of software updates and patches is decided based on version information, so the correspondence between the software installed on the devices and the version information of this software is the most important management target. Furthermore, unless cyber-attacks are carried out indiscriminately, an attacker will perform a preliminary investigation to find out the version information of the OS/software installed on the attack target device and then select an attack method to exploit the vulnerabilities that exist in these specific versions. Version information is essential for preparing for targeted attacks, narrowing down the means of attack in advance, and reducing risk.

In particular, when control systems are deployed for operation in the field, they are often not kept up to date like IT systems with the latest operating systems and security patches [28].

(2)  System password information

Malware is frequently created to infiltrate huge numbers of IoT devices with weak system passwords. These work by indiscriminately invading IoT devices using a list of insecure passwords that are easily guessed, such as 123456, qwerty, or password. IIoT devices in industrial and social infrastructure systems will probably be less exposed to this type of cyberattack than IoT devices that access the Internet. On the other hand, system password management is crucial for avoiding the risk of this type of malware spreading through internal networks. In particular, the sites where control systems are running do not always have in place the sorts of password policies that are becoming common in IT systems.

As described above, this paper proposes and evaluates an asset configuration management agent that can collect software version information and system password information with a low processing overhead. It is also considered that the proposed method can collect other kinds of information, such as (a) individual identification information and (b) network connection information of the abovementioned devices, which can be put to effective use in the future security updates of each device and automation of hardening measures.

*4.2. Setting Performance Targets for Asset Configuration Management Agents*

Performance targets are set for the asset configuration management agent to satisfy the requirements of the control system. In control devices, such as DCSs and PLCs, that are used in IIoT systems, and in some IoT devices, real-time control processing is needed to control plant equipment. In real-time control processing, inputs to the system require outputs to be generated within a specified response time.

Specifically, the response time is ensured by reliably running the control process within a prescribed control cycle of about 1 ms to 10 ms in duration. Depending on the speed of this control cycle, some devices may require a small amount of deviation (jitter) from the specified control cycle duration. Therefore, when the asset configuration management agent mounted in a control device or the like acquires asset configuration management data, such as software management information and password information, as described above, this must not hold up the real-time control process. We, therefore, set the following two items as performance targets for asset configuration management agents:

(1)   CPU load factor

In the CPU of a control unit, the control processes and security setting information acquisition processes are run in a time-sharing manner by the operating system. If an acquisition process occupies the CPU for too long, the start period of the control process will be disrupted, which could impair the control stability. Therefore, an upper limit is defined for the amount of CPU time that the acquisition process is allowed to take, and the system is designed so that processes cannot take more time than this.

The limit of the CPU load factor at which real-time control can be stably executed depends on the scheduling method. In a typical scheduling method called rate monotonic scheduling (RMS) [29], it has been shown that response times can be almost certainly guaranteed up to a load factor of about 80%. In the design of control processing using real-time operating systems in PLCs and the like, it has also been reported that a maximum CPU load factor of roughly 70–80% is an appropriate design criterion [30]. Therefore, if the CPU load factor of the acquisition process does not exceed about 10%, it can be installed on a control device without interfering with the control process. Since the CPU load may increase temporarily during actual operation, the target CPU load factor for acquisition processing is set to 1% or less, which is much smaller than the general margin of 10%.

(2)   Memory occupancy

If the data acquisition process occupies more of the control device's memory than necessary, there may not be enough memory available for the control processes to run properly, causing them to fail or run too slowly and damaging the control stability. Although control devices can be modeled uniformly with the granularity of Section 3.1 regardless of the type of control system, it is difficult to model them uniformly down to the detailed task level. In this paper, instead of discussing theoretically recommended memory occupancy values, we instead assume a maximum memory occupancy of about 90% as a general principle when designing control processes. To maintain the stability of the control processes, the memory occupancy rate of the acquisition process is designed to be no more than 1%, which is much smaller than the 10% margin.

### 4.3. Automating Asset Configuration Management

We propose a method for using an asset configuration management agent to automate the asset configuration management of IIoT systems with the configuration shown in Figure 4. For effective security measures, in addition to automatically collecting asset configuration management data, it is necessary to have a system that deploys patches and security settings to each asset in cooperation with systems, such as external servers, that manage vulnerabilities. It is also important to visualize the status of each device, and to generate a visual warning to urge system administrators to take action when a security vulnerability has been found in a device. Figure 5 shows the prototype GUI that we implemented for this paper. This is an example of a system that allows the administrator of an IIoT system to easily implement security measures. For example, it allows administrators to view the configuration of an IIoT system and the asset configuration management data collected by the asset configuration management agent. It can also identify the location of vulnerable devices.

The various control devices and IoT devices used in IIoT systems also run diverse operating systems and software environments. For example, while some devices may use TRON-based real-time operating systems, others may use highly versatile operating systems, such as Linux and Windows, depending on the application. Consequently, the types and data formats of the information acquired from these devices by the asset configuration management agent will also be diverse.

To collect and centrally manage the asset configuration management data, the asset configuration management agent should, therefore, be equipped with processing algorithms that are appropriate for each device and the type of network to which it is connected. For this purpose, we propose a design environment that uses Node-RED [31] in a server

that manages asset configuration management agents to facilitate the addition of processing algorithms compatible with various devices and networks. As shown in Figure 6, processing algorithms can be designed flexibly and easily by visually combining existing or new functional blocks.

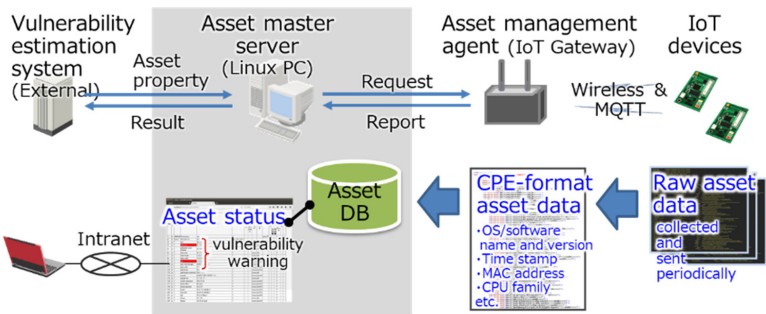

**Figure 4.** Architecture of asset configuration management system.

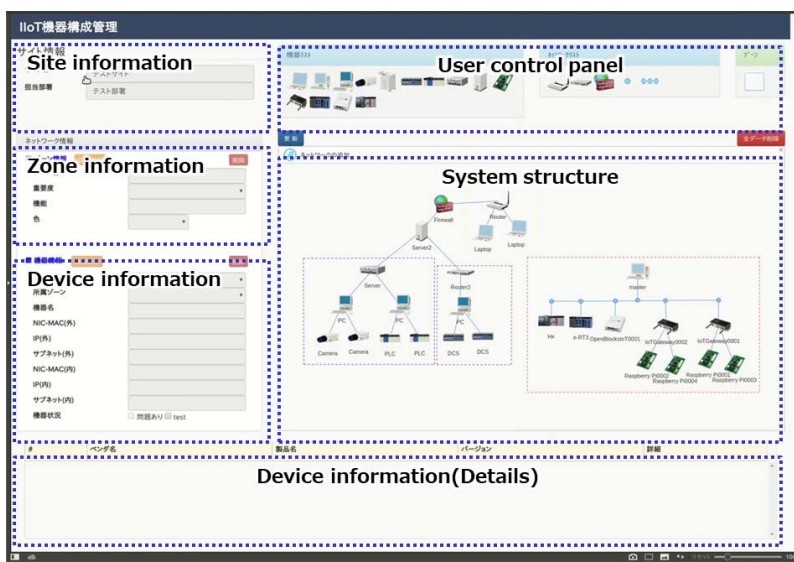

**Figure 5.** GUI example for asset configuration management.

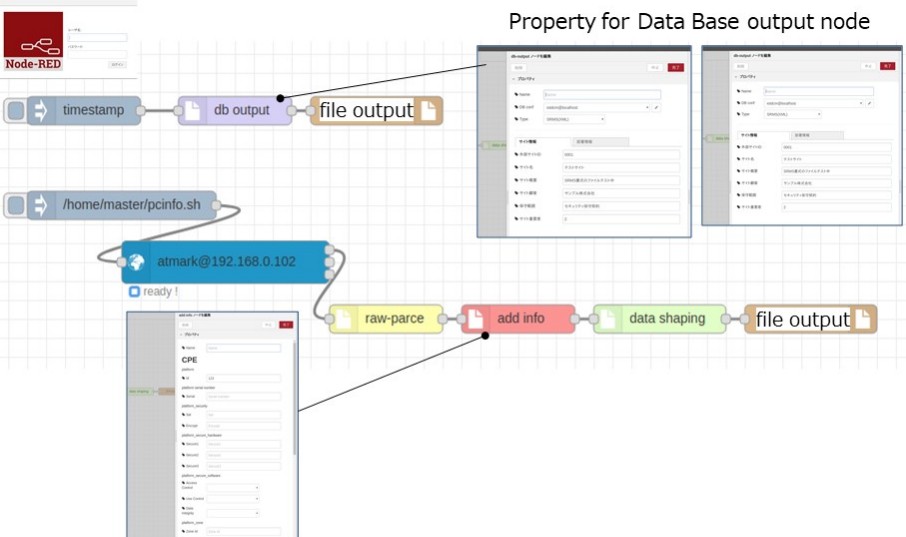

**Figure 6.** Algorithm design on Node-RED environment.

## 5. Prototyping and Evaluation of an Asset Configuration Management Agent

*5.1. Implementation Method*

In this section, we develop the prototype of asset configuration management agent that has functions to acquire version information about installed software and check system passwords. We implement the prototype to actual IIoT devices and evaluate the capability of the asset configuration management agent. We discuss the test results and confirm that our method can achieve the performance targets.

We selected the IIoT devices shown in Table 2 as targets to be equipped with the asset configuration management agent functions for collecting software version information and system password information, as described in the previous section. Since they are all running a Linux-based operating system, they can be compared under the same conditions when conducting the performance evaluation described below.

**Table 2.** IIoT devices for security information collection.

| Specification | PLC | IoT Gateway | IoT Device (Raspberry Pi 3) |
|---|---|---|---|
| CPU | ARM Cortex-A9 (32 bit dual, 866 MHz) | ARM Cortex-A7 (32 bit dual, 1.0 GHz) | ARM Cortex-A53 (64 bit quad, 1.4 GHz) |
| RAM | 256 MiB | 512 MiB | 1 GiB |
| OS | Linux 3.18.16-rt13 | Linux 4.9 | Raspbian 9 |

In the following, we describe the methods selected in Section 4.1 to collect software version information and system password information in asset configuration management agents.

(3)    Software version information collection function

In Linux for PCs and servers, tools (a) and (b) below are mainly used for the purpose of software version control. However, the versions of Linux used in embedded systems, such as those used for IIoT devices, may not have these tools due to storage capacity limitations. Therefore, the asset configuration management agent uses a more general method (c). This is a primitive approach that involves directly starting the managed software to obtain the version information. However, since it is supported by most of the officially distributed software, it is thought to be applicable to a wider range of devices and software.

(a)    Package management system (RPM for RedHat Linux, dpkg for Debian Linux, etc.)
(b)    Binary library management software (pkg-config, etc.)
(c)    Running software with a command line option to display version information (–version, -v, -V)

(4)    System password information checking function

To check for password vulnerabilities, the passwords recorded in the system password database (/etc/shadow) of the target device are read and checked against a list of known vulnerable passwords. Since the system passwords stored in/etc/shadow are hashed together with random data called "salt", the passwords are checked by hashing known weak passwords with the same salt values to see if any of them produce the same results. This method is highly secure because it is possible to inspect the system passwords stored in a target device without sending them over the network. However, since the computational load of this process increases as the number of system passwords and known passwords to be compared increases, the number of consecutive comparison processes is limited to suppress the increase of the CPU load.

To achieve the performance goals described in Section 4.2, we have drawn up design guidelines for the above two functions (1) and (2). So as not to overload the CPU, these functions are executed with the lowest priority, and the process sequences are split into

multiple subprocesses by calls to the "nanosleep" system call that allows them to be suspended. With this method, the CPU occupancy time of the asset configuration management agent can be reduced, thereby ensuring the responsiveness of control-related processes. Furthermore, to suppress the memory occupancy rate, the data buffers to be reserved in advance are made as small as possible, and only the minimum necessary data are fetched from files or networks for processing.

### 5.2. Performance Evaluation

To ensure that real-time control is not inhibited by the information collection function implemented in the agent, we used the following methods to evaluate the performance targets set in Section 4.2.

(1)    CPU load factor

Linux has a counter that records the cumulative CPU time expended on each process, which is used by the scheduler to evaluate the execution order of the processes. This cumulative value can be retrieved through an interface called the proc filesystem (/proc/«process ID»/stat). By reading this cumulative value at regular intervals and dividing the difference from the previous time by the read interval, it is possible to ascertain how the CPU load factor varies among the target processes. Normally, the load factor in a PC or server is measured at one second intervals, but, in this study, we decided to measure the CPU load factor at intervals of 10 ms, which is used as a general control cycle because it is not acceptable to disrupt real-time control even for a very short time.

(2)    Memory occupancy

The Linux kernel also manages the memory occupancy of each process, which can be ascertained through the same proc filesystem interface. Like the CPU load factor, the memory occupancy is also measured at 10 ms intervals to ascertain the tendency for the memory load factor to increase immediately after a process has been started up.

### 5.3. Discussion of Evaluation Results

Figures 7–9 show the results of evaluating the performance of the asset configuration management agent using the methods described in the previous section.

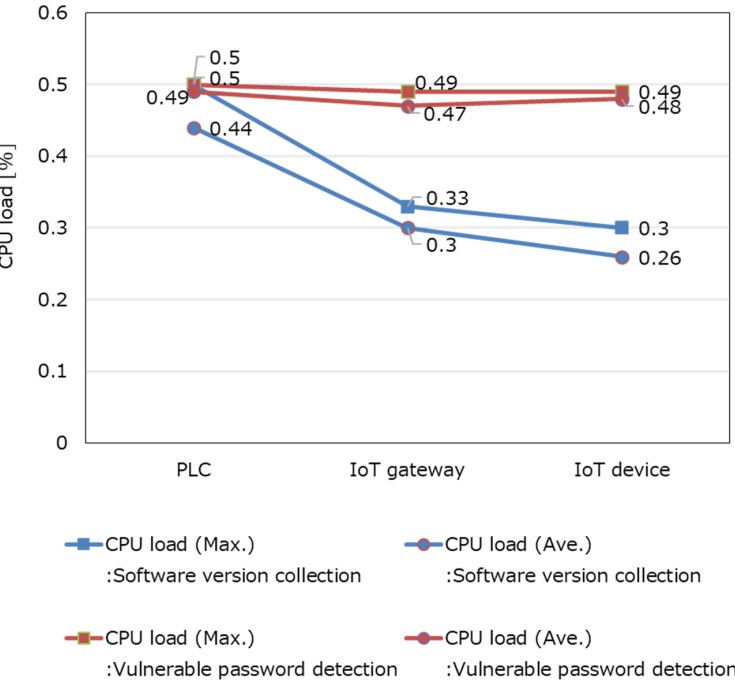

**Figure 7.** Performance test (CPU load).

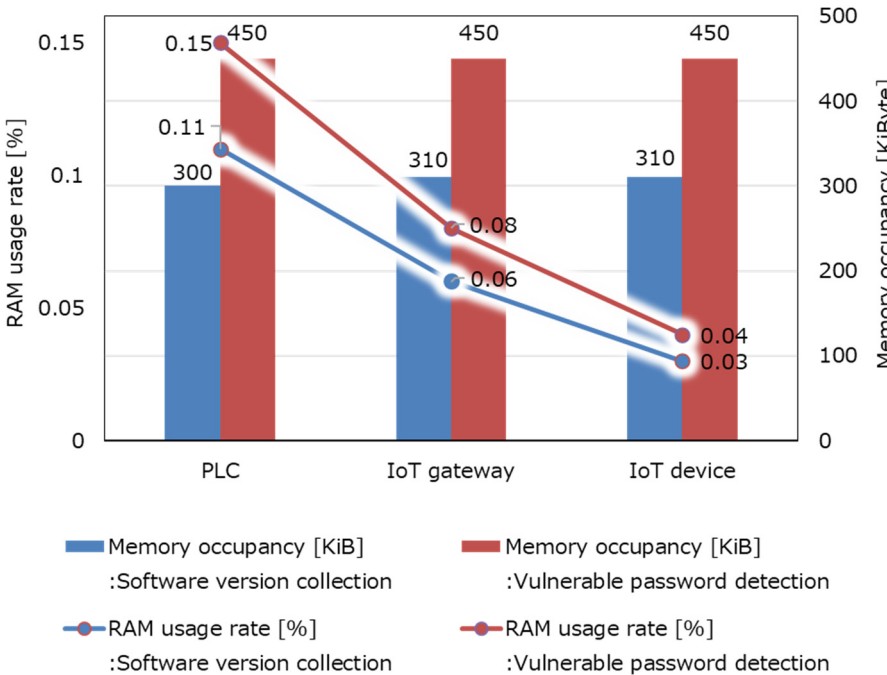

**Figure 8.** Performance test (memory consumption).

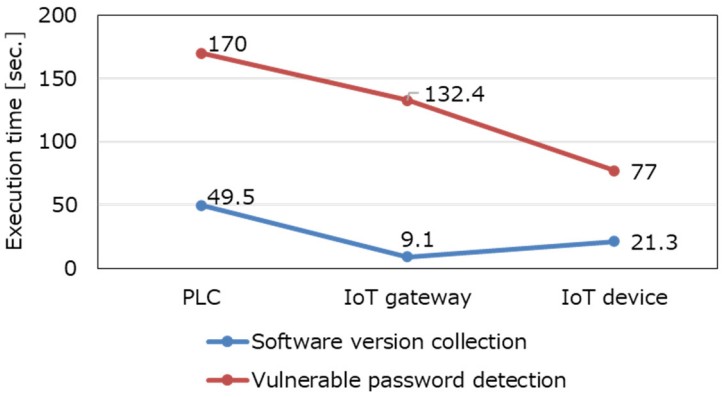

**Figure 9.** Performance test (execution time).

Both the CPU load ratio and memory occupancy results confirmed that the set performance targets were satisfied for all the target IIoT devices. In the collection of version information, the execution time can differ greatly from one device to the next, but this is due to differences in the number of installed applications. In the password checking process, the processing continues with an average processing load close to the maximum CPU load factor of about 0.5%. We think this is due to the processing load associated with password hashing.

In addition, in the comparison of the results for each IIoT device, since PLCs have inferior CPU performance (i.e., slower operating frequencies), their execution times are longer. On the other hand, their CPU load factors are still lower in the password checking process. We think this is because real-time patches are applied to their operating systems so that the processing load can be distributed more precisely than in other operating systems. In addition, since software updates generally occur about once a day at most, we believe that updates will not cause any practical difficulties if their execution time is within a few minutes.

We measured the execution time required for the password checking process, which was 170 s for the PLC, about 132 s for the IoT gateway, and about 77 s for the IoT device, as shown in Figure 9. Since the processing was divided to reduce the CPU load factor, the

execution time exceeded one minute. The recommended methods for securely setting and managing passwords are described in IEC62443 [32], which is an international standard for control system security. Although the expiration date of passwords is an important consideration from the viewpoint of human errors, such as reusing the same password or storing them insecurely, it is generally acceptable to continue using the same password for several months. Therefore, we believe that execution times of the order of minutes will not affect the practicality of this approach.

Although support for other operating systems is an issue for future work, we believe that the asset configuration management agent design method proposed in this paper can be applied to devices running any operating system.

## 6. Conclusions

We have proposed an agent-based approach to the automation of asset configuration management, which is important for deploying security measures in IIoT systems. We have shown that our proposed asset configuration management method can automate the understanding of the asset configurations in IIoT systems and the collection of information from these systems for the use in implementing security measures. We believe that the use of this method will make it possible to reduce the administrator workload needed for security countermeasures and will contribute to strengthening the security of industrial control systems that are being increasingly targeted at IoT applications.

Some future prospects include using automatically collected asset information as the basis for automating actual security measures, such as software security updates and the hardening of device settings. For practical deployments, it may also be possible to implement secure coding and communication encryption functions without increasing the processing load so that the asset configuration management agent itself does not become a security hole.

**Author Contributions:** Conceptualization, N.M., J.F. and H.E.; Investigation, N.M., J.F. and H.E.; methodology, N.M., H.E. and T.Y.; software, N.M. and H.E.; writing—original draft preparation, N.M. and H.E.; writing—review and editing, N.M., T.Y., K.S. and O.K; visualization, N.M. and H.E.; project administration, N.M.; supervision; T.Y., K.S. and O.K. All authors have read and agreed to the published version of the manuscript.

**Funding:** This research received no external funding.

**Institutional Review Board Statement:** Not applicable.

**Informed Consent Statement:** Not applicable.

**Data Availability Statement:** Not applicable.

**Conflicts of Interest:** The authors declare no conflict of interest.

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
