# Peer review of "Asset Management Method of Industrial IoT Systems for Cyber-Security Countermeasures"

_information, doi:10.3390/info12110460_

Round 1
Reviewer 1 Report
Line 28: IIoT technology uses a diverse variety of wired and wireless connection methods, which makes the network configuration more complex and makes it difficult to manage devices manually.
Recommend citing relevant papers such as:
Bartoli, A.; Dohler, M.; Kountouris, A.; Barthel, D. Advanced security taxonomy for machine-to-machine (M2M) communications in 5G capillary networks. In Machine-To-Machine (M2M) Communications; Woodhead Publishing: Sawston, UK, 2015; pp. 207–226.
Dhirani, L.L.; Armstrong, E.; Newe, T. Industrial IoT, Cyber Threats, and Standards Landscape: Evaluation and Roadmap. Sensors 2021, 21, 3901. https://doi.org/10.3390/s21113901
Line 34: missing Reference
Line 63: Is there any reference to back the comment made for legacy equipment?
Line 116: The scope of this paper is unclear. The authors are talking about IT/OT convergence and IoT based security issues and then introduce ISA95. Why isn’t ISA99 considered?
IEC62442 isn't discussed either which plays a huge role in terms of standardising IT/OT security and provides security controls for Asset Management.
Are you talking about connecting Legacy equipment with modern IT infrastructure? In that case many other security issues come into place.
And if you are considering an IIoT environment, you should consider ISA99 model (Level 0-5).
Reviewer 2 Report
The paper presents a high-level asset management system for cybersecurity countermeasure in an IIoT environment.
The abstract needs to clearly state the contributions of the paper and the kye results or highlights.
The introduction should briefly describe the key papers on the topic in one paragraph. These papers can be later analyzed in-depth in the Related Work section. The contributions should also be enumerated.
The related work section should be better organized and key papers on IoT automation and security are missing. Such papers should be discussed for completeness:
1. H Chegini et al. Process Automation in an IoT–Fog–Cloud Ecosystem: A Survey and Taxonomy, IoT 2 (1), 92-118
2. Towards Secure Fog Computing: A Survey on Trust Management, Privacy, Authentication, Threats and Access Control AAN Patwary et al. Electronics 10 (10), 1171 3. FogAuthChain: A secure location-based authentication scheme in fog computing environments using Blockchain AAN Patwary et al. Computer Communications 162, 212-224 4. A Framework of Automation on Context-Aware Internet of Things (IoT) Systems H Chegini, A Mahanti Proceedings of the 12th IEEE/ACM International Conference on Utility and Cloud Computing Companion 5. A Realistic and Efficient Real-time Plant Environment Simulator J Seo et al. International Symposium on Networks, Computers and Communications (ISNCC) Section 3 is very long. It would be good to break it into sections. At the start provide an overview of what is included in the section. Perhaps include a flowchart of the key components of the management system. Section 4 can have a few graphs to show the performance evaluation. It would be good to highlight the key results. The conclusions section is too long. It would be good to keep it short and focused. The figures can be made clearer. They are fuzzy and hard to read.Author Response
Please see the attachment.

Round 2
Reviewer 2 Report
Thanks for addressing all my feedback diligently. I have no further comments.